# Prediction of pharmaceutical and non-pharmaceutical expenditures associated with Diabetes Mellitus type II based on clinical risk

**Javier-Leonardo Gonzalez-Rodriguez**[1]*, **Carlos Franco**[1], **Olga Pinzón-Espitia**[2], **Vicent Caballer**[3], **Edgar Alfonso-Lizarazo**[4], **Vincent Augusto**[5]

**1** School of Management and Business, Universidad del Rosario, Bogotá, Colombia, **2** Facultad de Medicina, Departamento de Nutrición Humana, Universidad Nacional de Colombia, Hospital de la Misericordia, Universidad Del Rosario, Bogotá, Colombia, **3** Finanzas Empresariales, Universidad de Valencia, Valencia, Spain, **4** Université Jean Monnet Saint-Étienne, LASPI, Saint-Étienne, France, **5** Mines Saint-Etienne, Univ Clermont Auvergne INP Clermont Auvergne, CNRS, LIMOS Centre CIS, Saint-Etienne, France

* javier.gonzalez@urosario.edu.co

## Abstract

### Objective

To assess the effectiveness of different machine learning models in estimating the pharmaceutical and non-pharmaceutical expenditures associated with Diabetes Mellitus type II diagnosis, based on the clinical risk index determined by the analysis of comorbidities.

### Materials and methods

In this cross-sectional study, we have used data from 11,028 anonymized records of patients admitted to a high-complexity hospital in Bogota, Colombia between 2017–2019 with a primary diagnosis of Diabetes. These cases were classified according to Charlson's comorbidity index in several risk categories. The main variables analyzed in this study are hospitalization costs (which include pharmaceutical and non-pharmaceutical expenditures), age, gender, length of stay, medicines and services consumed, and comorbidities assessed by the Charlson's index. The model's dependent variable is expenditure (composed of pharmaceutical and non-pharmaceutical expenditures). Based on these variables, different machine learning models (Multivariate linear regression, Lasso model, and Neural Networks) were used to estimate the pharmaceutical and non-pharmaceutical expenditures associated with the clinical risk classification. To evaluate the performance of these models, different metrics were used: Mean Absolute Percentage Error (MAPE), Mean Squared Error (MSE), Root Mean Squared Error (RMSE), Mean Absolute Error (MAE), and Coefficient of Determination ($R^2$).

### Results

The results indicate that the Neural Networks model performed better in terms of accuracy in predicting pharmaceutical and non-pharmaceutical expenditures considering the clinical risk based on Charlson's comorbidity index. A deeper understanding and experimentation

**Data Availability Statement:** The datasets used for the current study are owned and managed by Hospital Méderi in Bogotá, Colombia. In this sense, we are not able to share any raw data without violating the terms of the ethics approvals. Hence, given the recommendation from Hospital Méderi Ethics Committee, data are not publicly available as they include sensitive and potentially identifying patient information. If there is anyone interested in request this information can contact to Luis Carlos Venegas, research director of Hospital Méderi (cimed@mederi.com).

**Funding:** The funders had no role in study design, data collection and analysis, decision to publish, or preparation of the manuscript.

**Competing interests:** The authors have declared that no competing interests exist.

with Neural Networks can improve these preliminary results, therefore we can also conclude that the main variables used and those that were proposed can be used as predictors for the medical expenditures of patients with diabetes type-II.

## Conclusions

With the increase of technology elements and tools, it is possible to build models that allow decision-makers in hospitals to improve the resource planning process given the accuracy obtained with the different models tested.

## Introduction

Diabetes mellitus (DM) describes a group of metabolic disorders characterized by high blood glucose levels. Worldwide, this illness is one of the most prevalent growing epidemics. It not only causes premature death and disability but also produces an economic loss, and is a significant threat to global development [1]. According to the International Diabetes Federation, [2] 463 million people had Diabetes Mellitus in 2019 (9.3% of the global population). Without sufficient action to address this epidemic, it is predicted that 578 million people (10.2% of the population) will have Diabetes by 2030. That number will jump to a staggering 700 million (10.9%) by 2045.

Diabetes is linked to macrovascular complications such as cardiovascular, cerebrovascular, and peripheral vascular diseases, and microvascular complications such as retinopathy, nephropathy, and neuropathy [1], as well as multimorbidity, i.e., the co-existence of two or more chronic conditions in the same person [3]. Despite the differences in population demographics and prevalence of these index conditions, there are common patterns concerning comorbidity, utilization, and costs. These common patterns could illustrate the underlying needs of people with multimorbidity that are often obscured in the literature that is still single-disease-focused [4].

The chronic nature of Diabetes and its related complications make it a costly disease, so, in order to provide some references for reducing this economic burden, it's important to investigate the factors influencing the hospitalization costs of patients with type 2 Diabetes [5]. DM, like most non-contagious chronic diseases, is associated with multimorbidity, but despite the recognized importance of the relationship between co-morbidities, the level of clinical risk of Diabetes, and the increased costs that this condition causes, there are not enough studies that examine this.

Although individual diseases dominate healthcare delivery, medical research, and medical education, people with multimorbidity need a broader approach [6]. This co-occurrence of diseases has implications from a disease management point of view, as the features of comorbid diseases can be much more complicated than a simple aggregation of individual illnesses [6]. Previous studies have related DM to a set of diseases such as cardiovascular, renal, obesity, and metabolic syndrome [7].

Due to the importance of Diabetes associated with an aging population and comorbidities as chronic diseases, it is necessary to develop predictive models to achieve adequate planning for the management of patients and the resulting costs. This requires, above all, the development of models for clinical risk assessment and classification of patients.

Comorbidity is associated with worse health outcomes, more complex clinical management, and increased healthcare costs [8]. It may be explained as a particular phenomenon of

interest within the domains of clinical care, epidemiology, or health services planning and financing. Mechanisms that may underlie the coexistence of two or more conditions in a patient (direct causation, associated risk factors, heterogeneity, independence) must be examined, and the implications for clinical care considered.

Morbidity scores, designed to summarize comorbidity for individual patients by summing scores for selected diseases, are widely used in research and service monitoring to adjust for baseline differences in patient groups or service providers [9]. As one of the most recognized, Charlson's index was developed to predict 1-year patient mortality using comorbidity data obtained from the review of hospital charts [10]. The validation cohort was 685 breast cancer patients at a Connecticut teaching hospital from 1962 to 1969. The final Charlson's index score was the sum of 19 predefined comorbidities that were assigned weights of 1, 2, 3, or 6. These weights were based on the magnitude of the adjusted relative risks associated with each comorbidity in a Cox proportional hazards regression model. At least 9 studies, representing more than 30 000 patients, have validated Charlson's index in a wide variety of diseases for numerous clinical outcomes [11]. The relative risks for each calculated from the proportional hazards model were used to create a single prognostic variable combining age and comorbidity that is indicative of subsequent risk [12]. Charlson's index is the most widely used method for predicting patient mortality based on comorbidity data [10].

Understanding all aspects of Diabetes treatment is hindered by the complexity of this chronic disease and its multifaceted complications and comorbidities, including social and financial impacts [13]. These approaches are often time- and cost-consuming and have frequently been supported by simulation models. Based on the previous concepts of comorbidity and risk classification, it is possible then to come closer to a prognosis of clinical risks based on predictive models that allow a forecast of the evolution of the severity of the disease and the associated higher healthcare costs.

Although the model proposed in this study is based on the comorbidity index proposed by the Charlson model, it is important to note that other studies have been developed, also with a predictive application, based on mathematical models, such as the one carried out in Qatar, using epidemiological and demographic data. A population-level compartmental mathematical model was constructed and applied to Qatar. The model was stratified according to sex, age group, risk factor status, and T2DM status, and was parameterized by nationally representative data [14].

There are also studies focused on determine throw the use of certain variables if a patient can be classified as diabetic or non-diabetic. In this sense, [15] tested different machine learning approaches over an Indian diabetic data set aiming to determine which of the 15 algorithms tested performs better over different evaluation metrics. Where authors found that logistic regression is the one that perform better over the other methods tested. In a similar sense, [16] propose the evaluation of different machine learning approaches over the same classification problem of diabetes, but the main idea is to evaluate the pros and cons of the use of these types of algorithms. In this sense, there are other types of approaches for the same problem that tries to improve the quality of the classification, in other words to improve the quality of the algorithm used, for example [17] propose a hybrid method which is trained over a real data set and results are compared with the state-of-the-art algorithms.

Also noteworthy is a predictive study conducted in Nepal focused on forecasting the amount and cost of medicine to treat type II diabetes mellitus using knowledge of medicine usage from a developed country. This consisted of modeling the financial burden of T2DM medicines by estimating the cost of medicines to treat all cases of T2DM in Nepal over three decades. The study was based on the prevalence of T2DM, Nepalese drug costs, and T2DM medicines based on a profile of Australia [18].

Furthermore, according to Goecks et al. [19], machine learning promises a future of rigorous outcomes-based medicine with detection, diagnosis, and treatment strategies that are continuously adapted to individual and environmental differences. It is expected that one of the applications of this model will be oriented to the effective control of both the main disease and the comorbidities simultaneously, in order to reduce clinical risk and the impact on costs.

Several clinical investigations use prognostic modeling techniques to identify the health status of diabetic patients and characterize patterns of progression. It is important to be able to predict future health status, especially to apply prevention and intervention strategies in prediabetic individuals. Additionally, the implementation of a software system using a machine learning approach is recommended to improve Diabetes management, as well as to demonstrate and evaluate its own effectiveness. These systems are generally incorporated into management systems that handle the various factors affecting the health of people with Diabetes by combining multiple artificial intelligence algorithms. However, the literature regarding studies aimed at predicting not only the risk of clinical evolution but also the economic impact of the disease is rather scarce [20, 21].

In this context, it is evident that a study area that applies mathematical algorithms to simulate Diabetes and its potential outcomes has gained increasing attention [14]. On the other hand, the interest of this article is not centered on the modeling applied to the evolution related to clinical management but on the expenses related to the age and the scorecard of clinical risk due to comorbidity.

Predicting healthcare expenditure according to morbidity gives rise to two methodological questions: how to measure multimorbidity and which predictive model to use. Regarding the former, a valid alternative is to use a risk adjustment system to determine the multimorbidity [22]. Machine-learning methods have gained considerable attention as novel strategies for capturing the interdependence of health variables when making predictions of complex health events [23]. However, it is not very common to find machine learning applications involving the clinical risk based on comorbidities and the pharmaceutical and non-pharmaceutical expenses associated with a chronic non-transmissible disease such as Diabetes. In this sense, the contribution of this work lies in the prediction of expenditures of patients with diabetes type II (divided into pharmaceutical and not pharmaceutical) to improve budgeting. This means that we're not trying to predict if a patient will suffer from a specific illness, otherwise a monetary value would be predicted. In our case, we have used specific information about patients and their medical condition, and we have proposed the use of the Charlson index to use the medical information to classify patients and improve the performance of prediction. Some limitation of this work lies in the availability of data given that we have worked with a hospital and its database but it can be improved with other databases from different types of patients.

## Materials and methods

The general methodology used in this study is described in "Fig 1". The first step consists in preparing the raw anonymized data received from the hospital databases. In the second step, the data was split into the training set (75%) and the test set (25%) to analyze the accuracy of the tested models. The training set was used in the third step to train three different models. In the fourth step, the different performance metrics were calculated for the three models. Once the models were tested and evaluated by the different performance metrics, an analysis to compare the obtained results was conducted.

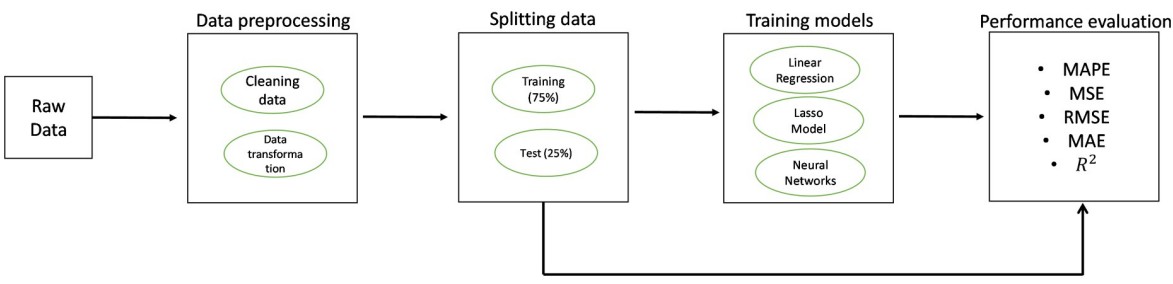

**Fig 1. Methodology of the study.**

## Data pre-datacasting

In this cross-sectional study, we have used data from anonymized records of patients admitted to a high-complexity hospital in Bogota Colombia between 2017–2019 with a primary diagnosis of Diabetes. The initial data set contains 11,028 records. Each time that a patient is admitted to the hospital is treated as a single source of data, therefore patients with multiple registrations to the hospital in different periods are analyzed as different records. Each anonymized record contains information related to the number of comorbidities detected, the number of treatments and medicines received, and the costs associated with medicines and medical services.

The data pre-processing step involves two main stages: cleaning data and data transformation. The initial database contains some missing values and outliers. Therefore, those outlier records and missing data (such as age >100 or expenses = 0) were deleted.

The data set was transformed to obtain the information for each anonymized patient (that was repeated in multiple records on the same date of the medical service delivery) into a single record. In this sense, the data were summarized obtaining a final database composed of 11,028 records where each patient can have different records, in other words, different consumptions of pharmaceutical and non-pharmaceutical expenditures. The variables for this study are summarized in Table 1. The first part of Table 1 shows the variables obtained directly from the

**Table 1. Variables used in the study.**

| | **ORIGINAL VARIABLES** | | |
|---|---|---|---|
| | Variable | Description | Type |
| Dependent Variables | Pharmaceutical expenditure | Pharmaceutical expenditure for each patient | Real ($) |
| | Non-pharmaceutical expenditure | Non-Pharmaceutical expenditure for each patient | Real ($) |
| Independent Variables | Visits to hospital | Number of times patient has been admitted to the hospital | Integer |
| | Sex | Sex of the patient | Binary (M/W) |
| | Age | Age of the patient | Integer (years) |
| | Length of hospital stay | Number of days at hospital for each patient | Integer (days) |
| | Comorbidities | Comorbidity(ies) detected for each patient when arrives to the hospital | Binary (type of comorbidity) |
| | **CALCULATED VARIABLES** | | |
| | Variable | Description | Type |
| Independent Variables | Pharmaceutical registers | Number of medicine consumptions for each patient | Integer |
| | Non-Pharmaceutical registers | Number of non-pharmaceutical consumption for eath patient | Integer |
| | Charlson's index per comorbidity | Identification for each comorbidity and its corresponding CHarlson's index | Integer |
| | Charlson's index for age | Identification of the Charlson's index for age | Integer |
| | Total Charlson's index | Determination of the Charlson's index for each patient | Integer |

original database. The second part of Table 1 shows the variables calculated based on the information from the original database.

As the main goal of this research is to evaluate a method to estimate the medical expenditure of patients with Diabetes and associated comorbidities, two dependent variables are used: pharmaceutical expenses and non-pharmaceutical expenses. As these dependent variables typically have distributions that show right-skewness with a large mass at zero [24], ordinary least squares (OLS) linear regression based on normal distribution has traditionally been used [25]. However, there is a risk of the error term of this model. For this reason, in this study, it has been decided to start from a classic model of multivariate linear regression along with other approaches that don't consider statistical assumptions such as Lasso and Neural Networks models to evaluate their performances. This is to identify the model that has the greatest predictive power, which can be used with greater accuracy in the planning of hospital expenses of patients with Diabetes and associated comorbidities. The independent variables are those which describe the characteristics of the patient (sex, age), the amount of medicine and services consumed (calculated with the database), and the associated comorbidities (calculated with Charlson's index).

In this study, Charlson's comorbidity index was chosen to classify each patient considering their comorbidities. This index developed by Charlson et al. [12, 26] is a validated method of classifying comorbidity to predict short- and long-term mortality from medical records. It replaces direct measures of the severity of an illness, which require prospective data collection. According to this index, we assigned weights for several major conditions present among secondary diagnoses. The next step was to calculate the total Charlson index. As explained in the methodology, this index is composed of two categories, one derived from the assignment of a value associated with the age group, which provides six levels. The other scale is assigned by the chronic diseases presented by the patient, which consists of four categories, which are added together to obtain the comorbidity index. Finally, the age indicator is added to the comorbidity indicator to obtain the total Charlson index.

In Table 2, we present the description of the data considering different factors such as the Charlson index, the sex, and the age of the patients. From this table, it can be concluded that if we group patients by the Charson's index, the average age also increases but most of the population is concentrated over indexes from 3 to 7.

## Training models

The following are the models used to predict the pharmaceutical and non-pharmaceutical expenditures associated with a Type II Diabetes Mellitus diagnosis, considering the clinical risk based on Charlson's comorbidity index.

### Multivariate linear regression

Multivariate linear regression is a statistical method to study the effect of several response variables and several predictor variables by calculating the coefficients predictors [27]. Firstly, the statistically significant variables were identified, and a predictive model of the hospital expenditure was composed considering both pharmaceutical and non-pharmaceutical expenditures. The Manning-Mullahy log [28] was used to determine the most appropriate statistical method and to estimate the distribution of the observations. As the coefficient of Kurtosis was greater than 3, linear regression using least squares was the recommendable statistical technique.

The dependent variable of the model was the log of healthcare expenditure. Transformation to a log was carried out since in this way the distribution of healthcare expenditure showed a better fit, for which the application of the Smearing Estimator was required [29].

**Table 2. Description of data by Charlson's Index distribution.**

| Charlson Index | | Count | % Rang | %Total | Age Average | Age Dev. |
|---|---|---|---|---|---|---|
| | | | Data description | | | |
| 0 | M | 138 | 79,8% | 1,57% | 32,62 | 7,30 |
| | F | 35 | 20,2% | | 39,26 | 12,06 |
| 1 | M | 185 | 56,6% | 2,96% | 42,14 | 11,87 |
| | F | 142 | 43,4% | | 43,82 | 11,33 |
| 2 | M | 380 | 51,4% | 6,70% | 52,03 | 10,83 |
| | F | 360 | 48,6% | | 51,29 | 11,54 |
| 3 | M | 645 | 51,0% | 11,45% | 61,21 | 8,89 |
| | F | 619 | 49,0% | | 60,57 | 8,70 |
| 4 | M | 1021 | 54,3% | 17,05% | 67,20 | 9,53 |
| | F | 861 | 45,7% | | 66,23 | 9,92 |
| 5 | M | 1166 | 55,1% | 19,18% | 73,49 | 8,56 |
| | F | 951 | 44,9% | | 71,58 | 8,77 |
| 6 | M | 1018 | 53,5% | 17,24% | 76,97 | 8,75 |
| | F | 885 | 46,5% | | 75,07 | 9,06 |
| 7 | M | 690 | 55,5% | 11,26% | 79,97 | 8,14 |
| | F | 553 | 44,5% | | 77,45 | 8,86 |
| 8 | M | 409 | 57,3% | 6,47% | 82,52 | 8,03 |
| | F | 305 | 42,7% | | 79,21 | 9,88 |
| 9 | M | 184 | 54,6% | 3,05% | 83,67 | 8,58 |
| | F | 153 | 45,4% | | 81,74 | 9,02 |
| 10 | M | 80 | 54,4% | 1,33% | 78,91 | 9,56 |
| | F | 67 | 45,6% | | 79,48 | 7,84 |
| 11 | M | 37 | 49,3% | 0,68% | 83,38 | 7,47 |
| | F | 38 | 50,7% | | 78,79 | 9,09 |
| 12 | M | 28 | 65,1% | 0,39% | 78,39 | 7,34 |
| | F | 15 | 34,9% | | 79,00 | 6,62 |
| 13 | M | 13 | 48,1% | 0,24% | 83,54 | 4,67 |
| | F | 14 | 51,9% | | 75,43 | 17,26 |
| 14 | M | 2 | 22,2% | 0,08% | 87,00 | 1,41 |
| | F | 7 | 77,8% | | 73,29 | 10,83 |
| 15 | M | 2 | 22,2% | 0,08% | 65,00 | 0,00 |
| | F | 7 | 77,8% | | 70,57 | 14,46 |
| 16 | M | 2 | 16,7% | 0,11% | 70,00 | 5,66 |
| | F | 10 | 83,3% | | 71,80 | 1,93 |
| 17 | M | 4 | 40,0% | 0,09% | 72,50 | 3,70 |
| | F | 6 | 60,0% | | 75,67 | 7,76 |
| 18 | M | 1 | 16,7% | 0,05% | 71,00 | 0,00 |
| | F | 5 | 83,3% | | 81,60 | 12,14 |
| 19 | M | 1 | 50,0% | 0,02% | 85,00 | 0,00 |
| | F | 1 | 50,0% | | 93,00 | 0,00 |

## Lasso model

The Lasso model (Least Absolute Shrinkage and Selection Operator) is a regression method developed to prioritize the importance of independent variables which minimizes the residual sum of squares and guarantees that the sum of the absolute value of the coefficients is less than

a predefined value [25]. The main idea of the Lasso model is to transform each coefficient by a constant obtaining improvements in prediction accuracy and interpretability. The experimental results show that this model works better than other methods by shrinking the coefficients exactly to zero. Hence, the Lasso model can be applied as an alternative to the conventional feature selection methods. It would be very useful for research groups, especially for those working on machine learning tasks, such as the one carried out and presented in this article [30].

## Neural networks

The Neural networks model was created to simulate the human brain and use the information to classify or predict data, the main idea is to build a neural network composed of neurons represented by nodes and arcs that in turn represent the connections between neurons [26]. The main idea is to find the weight of the arcs by using mathematical functions that converge via iterative calculations.

## Performance evaluation

The following measures were used to evaluate the performance of the models used to predict the pharmaceutical and non-pharmaceutical expenditures of patients with Diabetes type-2 and associated comorbidities, these metrics were analyzed from previous related studies [31]:

- Mean Absolute Percentage Error (MAPE): is the mean of the absolute percentage errors of prediction

$$MAPE = \frac{100}{n} * \sum_{i=1}^{n} \left| \frac{x_i - f_i}{x_i} \right|$$

- Mean Squared Error (MSE): is a measure of the mean squared differences of the errors.

$$MSE \frac{1}{n} * \sum_{i=1}^{n} (x_i - f_i)^2$$

- Root Mean Squared Error (RMSE): is a measure of the differences between the predicted values of a model and the observed values that are considered as the standard deviation of the residuals.

$$the\ RMSE = \sqrt{\frac{1}{n} * \sum_{i=1}^{n} (x_i - f_i)^2}$$

- Mean Absolute Error (MAE): the average distance between each observed value and the predicted value.

$$MAE = \frac{100}{n} * \sum_{i=1}^{n} |x_i - f_i|$$

- Coefficient of Determination ($R^2$): is a measure of how close the data and the predicted values are.

  Where $x_i$ and $f_i$ corresponds to the observed and predicted values respectively.

## Ethical considerations

This study conducted under the ethical standards based on the latest version of the Declaration of Helsinki, taking as a basis in clinical research the ethical principle of autonomy, non-maleficence, beneficence and justice of persons.

We also took into consideration Resolution 8430 of 1993, issued by the Colombian Ministry of Health, which establishes scientific, technical and administrative standards for health research [32]. It also considers the declaration of Taipei (WMA, 2002) regarding the use of massive health databases [33]. The research had the written approval of the Research Committee of the Mederi University Hospital of Bogota. Written approval was also obtained from the Ethics Committee of the Universidad Del Rosario, as required by the Hospital's Research Committee.

The Mederi Hospital provided the study databases from the information systems supplied by the hospital itself through the statistics and business intelligence department. The DBs did not contain personally identifiable information. In addition to which we should ensure full anonymization of the Dbs.

In addition to the above, we agreed that once the analysis of the information was completed, we would return the databases and delete them from the respective records. On the other hand, the Ethics Committee did not require informed consent from patients, since it was a research study without risk for individuals, but also because it was a retrospective study on the population selected for this purpose within the hospital, which is why it was logistically unaffordable to request consent from all participating individuals.

We also consider that we did not have personal contact with any person registered in the databases, since we only reviewed the information available in the records. We reviewed data on adult patients who hospitalized at the Mederi Hospital prior to the analysis of the information. The databases did not involve minors.

## Results

### Descriptive analysis

11,028 descriptive cases with a diagnosis of Type II Diabetes were studied, the highest proportion corresponds to the female gender, with 54%, equivalent to a total of 6000 cases. Age was classified according to Charlson's index into six groups (for each decade after the age of 50 a point is added up to 5 points): 0–49 years, 50–59 years, 60–69 years, 70–79 years, 80–89 years, and 90–99 years. According to this classification, 70% of the population studied is over 50 and the groups between 70 and 89 years of age represent 40% of the total. On the other hand, the characteristics of the pharmaceutical and non-pharmaceutical expenditures are as follows: the mean values are $664.94 and $1256,30 respectively, and the standard deviations are 1,756.03 and 2,720.07 respectively. Finally, there are records with no consumption (it means cost = 0) and the maximum values are $60,984.87 and $50,741.03 respectively.

According to Table 3, regarding Charlson's index distribution, the largest proportion of cases are in category 0 (46.5%), i.e., not classified in Charlson's index as high risk of mortality within one year. This is followed closely by the Charlson index's first category (45.2%), which includes comorbidities such as Myocardial Infarction, heart failure, Peripheral Vascular Disease, Dementia, Cerebrovascular Disease, chronic lung disease, connective tissue diseases,

**Table 3. Results of linear regression.**

| Variable | Coeff |
|---|---|
| Constant | 8.7121 |
| Pharmaceutical registers | 3.5092 |
| Non-pharmaceutical registers | -0.9120 |
| # of visits | -13.3634 |
| M | 36.8740 |
| Age | -0.6752 |
| Length of hospital stay | 6.8228 |
| Charlson Index Age | 13.5392 |
| Charlson Index | -32.2971 |

uncomplicated Diabetes, ulcers, and chronic liver disease. Charlson's second category corresponds to 6.9% of cases and gathers the comorbidities of higher clinical risk as Hemiplegia, chronic renal disease, Diabetes with complications, tumors, Leukemia, and Lymphoma. The remaining 1.31% of the cases corresponded to Charlson's sixth category, which includes Metastatic Neoplasms and AIDS. In Charlson's fourth category, corresponding to moderate or severe liver disease, no cases were recorded.

The total Charlson score is obtained from the sum of the indicator assigned by age group and the indicator corresponding to the associated comorbidities. 24% of cases remain unclassified because they present illnesses that do not correspond to the categories established by the Charlson index, such as acute or infectious diseases, given that the index only applies to chronic diseases. This score allowed us to classify the clinical risk into low, moderate, and high. The results obtained from Charlson's score are as follows: 3(11%), 4(17,4%), 5(19%), 6 (17,2%), 7(11,2%) and 8(24%). According to this score, 5, 6, and 7 account for 47.7% of the cases, which corresponds to medium and high clinical risk, considering age and comorbidities.

On the other hand, the most frequent hospital length stay corresponds to a stay between 1 and 7 days (50.4% of the cases), followed by a stay of up to 14 days (16.2% of the cases). According to this, 66% of the cases had a hospital stay between 1 and 14 days.

## Model results

The results of the models and their performance metrics are presented in two different parts: (i) Results for pharmaceutical expenditures (Tables 3–5), and (ii) Results for non-pharmaceutical expenditures (Tables 6–8).

### Pharmaceutical expenditures

As shown in Table 5, the Neural Network model has the best performance on the MAPE measures, improving this indicator by approximately 95% compared to the Lasso model. The Linear Regression model has the worst performance on this measure. The reader should note that, in the Linear Regression model, binary variable F is not included in order to guarantee the perfect collinearity assumption of the OLS.

In terms of the Mean Squared Error (MSE), the results are not sparse between the three models, the difference between the maximum and the minimum value is not higher than 20%. The Lasso model and the Neural Network model present the best performances compared with the Linear Regression model. The best performance is obtained by the Lasso model but

**Table 4. Results of the lasso model.**

| Variable | Coeff |
|---|---|
| Pharmaceutical registers | 1.35663793 |
| Non-pharmaceutical registers | -2.3791581 |
| # of visits | -28.797945 |
| M | 47.75424712 |
| W | 0 |
| Age | -1.09833083 |
| Length of hospital stay | -6.22483504 |
| Charlson Index Age | 24.85700929 |
| Charlson Index | -27.8375489 |

**Table 5. Performance indicators for the three analyzed models.**

| Performance Indicator | Linear Regression | Lasso Model | Neural Network |
|---|---|---|---|
| MAPE | 3.705,30 | 409.902968 | 209.791 |
| MSE | 1.26E+06 | 1.053067e+06 | 1086892.9757 |
| RMSE | 1.125,01 | 1026.1906 | 1042.5415 |
| MAE | 416.434 | 409.902968 | 385.1069 |
| $R^2$ | 0.622 | 0.619 | 0.7567 |

**Table 6. Results of linear regression.**

| Variable | Coeff |
|---|---|
| Constant | 206.6210 |
| Pharmaceutical registers | -0.3311 |
| Non-pharmaceutical registers | 5.0471 |
| # of visits | 26.2494 |
| M | 49.6800 |
| Age | -7.6726 |
| Length of hospital stay | 7.5528 |
| Charlson Index Age | -18.5200 |
| Charlson Index | 28.8678 |

**Table 7. Results of the lasso model.**

| Variable | Coeff |
|---|---|
| Pharmaceutical registers | 6.1748037 |
| Non-pharmaceutical registers | -1.93467062 |
| # of visits | 4.52867925 |
| M | 47.59657771 |
| W | 52.44208467 |
| Age | -0. |
| Length of hospital stay | -9.20765865 |
| Charlson's Index Age | 1.37978367 |
| Charlson's Index | 29.52544397 |

**Table 8. Performance indicators for the three analyzed models.**

| Performance Indicator | Linear Regression | Lasso Model | Neural Network |
|---|---|---|---|
| MAPE | 286.408581 | 234.132979 | 85.9203 |
| MSE | 1.469848e+06 | 1.522053e+06 | 1425399.8533 |
| RMSE | 1212.3730 | 1233.7152569889831 | 1193.9011 |
| MAE | 525.052057 | 539.851163 | 497.5700 |
| $R^2$ | 0.801 | 0.8039 | 0.8999 |

the difference with the Neural Network model is around 3% approximately. Similar behavior can be described with the RSME where the worst result is reached by the Linear Regression Model, and the best performance is obtained by the Lasso Model. In the case of the RSME measure, the difference between the maximum and minimum values is less than 10% and the difference between the Lasso Model and the Neural Network is less than 2%.

In terms of the Mean Absolute Error (MAE) measure, the difference between the limit values is approximately 8%, in this case, the Neural Network model presents the best performance result. Finally, in terms of the R-Squared metric, which can be analyzed as the percentage of variance explained by the dependent variables considered in the model, the Neural Network model presents the best overall performance.

## Non-pharmaceutical expenditures

As shown in Table 8, a similar conclusion to that obtained regarding the pharmaceutical expenditures can be described with the MAPE indicator, where the best result is reached by the Neural Network model and the worst result is obtained by the Linear Regression model; the results of the Neural Network outperform the results of the Linear Regression by 70%. As for the rest of the metrics, MSE, RMSE, MAE, and R-squared, the best results were obtained by the Neural Network model, which means that this model can be considered a good predictor of non-pharmaceutical expenditures.

## Analysis of results

In terms of applicability, we analyze each one of the results obtained by each model as follows:

### Linear regression model

In terms of the Mean Squared Error, the results show that the squared distance between the predicted values and the real values are 1.27E+06 and 1.469848e+06 for the pharmaceutical and non-pharmaceutical expenditures respectively. This metric is rather high given the wide range of pharmaceutical and non-pharmaceutical expenditures. Regarding the Root Mean Squared Error, the prediction model on average has an error of approximately US$ 1,125 and US$ 1.234 respectively. In terms of the Mean Absolute Error measure, the mean differences between the observations and the predicted values are US$ 416 and US$ 525 respectively. Finally, the R-Squared measure shows that the fitting for non-pharmaceutical expenditures (0.801) is better than the one obtained for pharmaceutical expenditures (0.622).

### Lasso model

For the Lasso model, the squared errors obtained for the pharmaceutical and non-pharmaceutical expenditures are 1.053067e+06 and 1.469848e+06 respectively, which improves the performance of the multivariate linear regression results. Nevertheless, this measure is still high

given the high variability of the input parameters. On the other hand, the RMSE measure is also improved compared with the Linear Regression Model, with an error of approximately US$ 1,026 and US$ 1,212 respectively. In terms of the MAE measure, this model provides an error of US$ 409 and US$ 539 respectively. Finally, the R-Squared measure shows once more that the non-pharmaceutical expenditure obtains a better performance (0.80) compared with the pharmaceutical expenditure (0.75).

## Neural network model

Finally, the results obtained by the Neural Network model improve some results obtained by the previously analyzed models. In terms of the MSE measure, the results are 1086892.97 and 1425399.85 for the pharmaceutical and non-pharmaceutical expenditures respectively. For the RMSE measure, the errors are US$ 1,212.3730 and US$ 1,193.9011 respectively, which improves the results for the non-pharmaceutical expenditures obtained by the Lasso model. In terms of MAE, the results obtained are US$ 525 and US$ 497 respectively, once more showing an improvement for the non-pharmaceutical expenditure. Finally, the R-Squared measure for both models improve those obtained by the other two models analyzed with respective values of 0.75 and 0.89.

## Discussion

Based on the results of this research, it is important to highlight that the classification of clinical risk based on comorbidity is a step that should be routinely assumed in the study of patients with chronic diseases, such as Type II Diabetes Mellitus. In this sense, it is essential to highlight that Charlson's comorbidity index is a good alternative to characterize comorbidities because it does not have large computer requirements and can be added to the patient's database at the time of reviewing the data or upon admission to the hospital. This can be structured in Excel or other similar formats, but the amount of data has to be analyzed before in order to use the correct database platforms. This process allows a risk classification from the time of admission and can thus be applied to the prediction of expenditure.

On the other hand, this study has shown that the use of machine learning models is a useful tool for predicting pharmaceutical and non-pharmaceutical expenditures. This fact is compatible with a previous study made in Pakistan [34, 35], where, by using a DELM approach, a high level of reliability with a minimum error rate was achieved. The approach shows significant improvement in results compared to previous investigations. "It is observed that during the investigation the proposed approach has the highest accuracy rate of 92.8% with 70% of training (9500 samples) and 30% of test and validation (4500 examples). Simulation results validate the prediction effectiveness of the proposed scheme" [35]. This allows us to think that predictive studies have better results with larger sample sizes, we have observed that when running some models in machine learning, with smaller sample sizes, we do not obtain conclusive and highly reliable results.

In another study in Pima India [36], a dataset of patients classified into diabetic and non-diabetic was analyzed to identify trends and detect patterns of morbidity and mortality associated with risk factors using R software. This contrasts with our study, in which we have gone further by developing and analyzing five different predictive models that have been coded in Python. This study has achieved outstanding results, their SVM-linear model shows exceptionally high accuracy of 0.89 and precision of 0.88 for the prediction of Diabetes compared to other models used, these were similar findings to those of our study [19].

There is a need for practical and effective predictive models to calculate and project future health expenditures. However, this task is difficult since the state of health is determined not

only by the complications of the basic disease but also, in the chronic and generally progressive types of Diabetes, by the pathologies that are associated concomitantly, known as comorbidities. Additionally, the variation of the costs derived from the health condition also contributes to making the projection of future health expenditures a difficult variable to calculate. Modeling studies may be useful in the future, including for replanning the diet of patients, as a way to control the determinants of disease [37].

The direct predecessors of this study were: "Predicting Healthcare expenditure by Multimorbidity Groups" [28], and "Modeling of Pharmaceutical Prescription Expenditure from CRG Stratification", both carried out in the Valencia Community [38]. The main contribution of the present work is in analyzing not only the classification of risk based on Charlson's comorbidity index but also the prediction of the expected cost based on the described association.

We consider that the use of the Charlson index to calculate the risk classification by comorbidity is a viable option for application in Colombia. The main reason for this consideration is that, in contrast to the CRG Clinical risk groups option, Charlson's index does not require the very robust systems and processors for information management and the fully electronic medical records that Colombia currently does not have [39].

## Conclusions

We have used different models to predict the relationship between the risk determined by Charlson's comorbidity index in diabetic patients, and the expected cost, both for pharmaceutical and non-pharmaceutical expenditures including different types of variables that describe the patients. Our database consisted of 11,028 records of patients admitted to a high-complexity hospital in Bogota, Colombia between 2017 and 2019 with a primary diagnosis of Diabetes. These cases were classified according to Charlson's comorbidity index in several risk categories, numerically determined by severity and age, with this variable proposed we were able to add a new variable that allows us to understand the medical condition of each patient and to improve the performance of the prediction.

Subsequently, from the determination of main variables, three machine models were used to predict hospital expenditure involving clinical risk classification. These models were: the Multivariate Linear Regression, the Lasso model (Least Absolute Shrinkage and Selection Operator), and Neural Networks. These models were tested and analyzed to evaluate the performance of different metrics or indicators of the errors obtained for each one of them, hence we have analyzed and compared the results contrasting these metrics that allow us to characterize and propose the best model.

The results indicate that Neural Networks performed better in terms of accuracy and prediction errors compared to regression models. This allows us to conclude that a deeper understanding and experimentation with Neural Networks can improve these first results. Therefore, from the managerial insights we can conclude that with this type of model and the variables proposed managers can improve the decision-making in terms of budget for the treatments of patients.

Diabetes management presents many opportunities for clinicians and artificial intelligence researchers to collaborate on challenging research that could potentially improve health, safety, and quality of life for millions of people living with Diabetes.

## Acknowledgments

The authors thank their colleagues at Méderi Hospital, Universidad del Rosario, Universidad Politécnica de Valencia, and Ecole de Mines Saint Etienne for their support with the data analysis.

## Author Contributions

**Conceptualization:** Javier-Leonardo Gonzalez-Rodriguez.

**Data curation:** Carlos Franco, Vicent Caballer.

**Formal analysis:** Carlos Franco, Vicent Caballer.

**Investigation:** Olga Pinzón-Espitia.

**Methodology:** Vincent Augusto.

**Supervision:** Olga Pinzón-Espitia, Edgar Alfonso-Lizarazo, Vincent Augusto.

**Validation:** Edgar Alfonso-Lizarazo.

**Writing – original draft:** Javier-Leonardo Gonzalez-Rodriguez.

**Writing – review & editing:** Javier-Leonardo Gonzalez-Rodriguez.

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
