## [Decision Letter · Decision Letter 0]

17 Jan 2023

PONE-D-22-29157Prediction of pharmaceutical and non-pharmaceutical expenditures associated with Diabetes Mellitus type II based on clinical riskPLOS ONE

Dear Dr. González R,

Thank you for submitting your manuscript to PLOS ONE. After careful consideration, we feel that it has merit but does not fully meet PLOS ONE’s publication criteria as it currently stands. Therefore, we invite you to submit a revised version of the manuscript that addresses the points raised during the review process.

We look forward to receiving your revised manuscript.

Kind regards,

Bruno Miguel Pinto Damásio

Academic Editor

PLOS ONE

Journal Requirements:

Additional Editor Comments (if provided):

Dear Dr Javier González,

I am writing to you regarding Manuscript Number PONE-D-22-29157, Prediction of pharmaceutical and non-pharmaceutical expenditures associated with Diabetes Mellitus type II based on clinical risk.

We have now received the referee reports for your article, and in light of them unfortunately the article cannot be accepted for publication in PLOS ONE in its current state.

One referee recommends rejection, two recommend a major revision.

Nevertheless, I invite you to respond point by point to the referees, as well as to improve your paper substantially, taking the comments into account.

Kind regards,

Bruno Damásio.

Reviewers' comments:

Reviewer's Responses to Questions

**Comments to the Author**

1. Is the manuscript technically sound, and do the data support the conclusions?

Reviewer #1: Partly

Reviewer #2: Partly

Reviewer #3: Partly

2. Has the statistical analysis been performed appropriately and rigorously? 

Reviewer #1: N/A

Reviewer #2: Yes

Reviewer #3: No

3. Have the authors made all data underlying the findings in their manuscript fully available?

Reviewer #1: No

Reviewer #2: No

Reviewer #3: No

4. Is the manuscript presented in an intelligible fashion and written in standard English?

Reviewer #1: No

Reviewer #2: Yes

Reviewer #3: Yes

5. Review Comments to the Author

Reviewer #1: The context of this study has a high relevance and can potentially be useful for resource planning in hospitals.

However, before publication, several issues must be addressed.

Firstly, the conclusions of this study are not yet compelling enough for the analysis done. In that sense, I ask you to rewrite that section (as well as the abstract).

This is important because you mention the relevance of studying the relationship between the input-output variables as a motivation for the study. Tables 3 and 4 are quite interesting and go in line with the initially mentioned matter of the paper. However, there is no actual discussion on this.

A Reorganisation of the document must be done. For instance, sections of the discussion clearly belong in the introduction or related work. Figure 1 doesn't need to be segregated from the main text. Related studies and descriptions of the previously used methods are part of the related work, not the discussion. The first part of the manuscript seems to be stitched together without much consideration for the previous methodological work. Then the second part mentions some previous work's results, but the discussion needs to be completed.

The lack of reference to important studies in this context shows a lack of a comprehensive review, which leads to some bold claims. For instance, how does this research build upon:

Awad, Susanne F., et al. "Forecasting the burden of type 2 diabetes mellitus in Qatar to 2050: a novel modeling approach." Diabetes research and clinical practice 137 (2018): 100-108.

or

Khanal, Saval, et al. "Forecasting the amount and cost of medicine to treat type 2 diabetes mellitus in Nepal using knowledge on medicine usage from a developed country." Journal of Pharmaceutical Health Services Research 10.1 (2019): 91-99.

Please also review the English and eventual translations like "Data transformación" and "Data Pre-Datacasting". Check also the comas and points issue: "US$ 1,125 and US$ 1.234"

Also, format the tables in the same fashion throughout the text for uniformity.

There must be at least a reference to support the choice of performance evaluation. A few graphs on the Descriptive Analysis are a critical addition to better categorize de distributions.

For the above-mentioned reasons, I believe that this manuscript has some potential but needs a comprehensive amount of work before publication.

Reviewer #2: Summary

The goal of this work is to assess the effectiveness of different machine learning models in estimating pharmaceutical and non-pharmaceutical expenditures associated with Diabetes, considering the comorbidities of the patient. The models considered were Linear Regression, Lasso Regression, and Neural Networks. The results showed greater accuracy in predicting pharmaceutical and non-pharmaceutical expenditures through Neural Networks.

Minor comments

The submitted document had two different formatting types. In future submissions, try to uniformize the font and size used.

In table 3, the results of the linear regression display the estimation of the intercept (constant), the dummy variable for Man (M), and the dummy variable for Woman (W). This table should be reviewed, since in the linear model, including both categories and the intercept would mean we are violating the absence of perfect collinearity assumption, which means we could not estimate the model through OLS.

More details on the methods used should be added in the Methodology and Results section. The results could also be complemented with some plots.

Major comments

Overall, this work is well written. However, it falls short concerning the implemented methodology. The result that Neural Networks are more efficient is expected. These algorithms have been used before to predict not only the risk/probability of a specific disease but also medical costs. The major contribution is the inclusion of Charle's Index as an attribute in the models. Thus, this work should be enriched with other methods, besides the ones implemented. For example, cluster the individuals based on the Charle's Index and then apply the models to each cluster or consider the panel data structure (since in this work each time that a patient is admitted to the hospital is treated as a single source of data, therefore patients with multiple registrations to the hospital in different periods are analyzed as different record).

Reviewer #3: The authors need to explain indepth all the used variables in the model. Note that using the same variable as independent and dependent invalidates any accuracy of the model, hence the authors must strongly justify why they used hospitalization costs as independent variable (which include pharmaceutical and non-pharmaceutical expenditures), and as dependent variable also expenditure (composed of pharmaceutical and non-pharmaceutical expenditures). This intorduces a high risk of biased resuts.

The authors must also rewrite the introduction and inclute the study contributions, as well in the methododology, it is fundamental to support theoretically each used item/variable included in the model. It is of most importance the conclusions contain implications and limitations of the study.

6. PLOS authors have the option to publish the peer review history of their article (what does this mean?). If published, this will include your full peer review and any attached files.

Reviewer #1: No

Reviewer #2: No

Reviewer #3: No

---

## [Author Response · Author response to Decision Letter 0]

13 Apr 2023

Reviewer #1: The context of this study has a high relevance and can potentially be useful for resource planning in hospitals.

However, before publication, several issues must be addressed.

• Firstly, the conclusions of this study are not yet compelling enough for the analysis done. In that sense, I ask you to rewrite that section (as well as the abstract).

This is important because you mention the relevance of studying the relationship between the input-output variables as a motivation for the study. Tables 3 and 4 are quite interesting and go in line with the initially mentioned matter of the paper. However, there is no actual discussion on this.

R:// We appreciate the comment and have improved the introduction, abstract, and conclusions clarifying the contribution and limitations of the study.

• A Reorganisation of the document must be done. For instance, sections of the discussion clearly belong in the introduction or related work. Figure 1 doesn't need to be segregated from the main text. Related studies and descriptions of the previously used methods are part of the related work, not the discussion. The first part of the manuscript seems to be stitched together without much consideration for the previous methodological work. Then the second part mentions some previous work's results, but the discussion needs to be completed.

R:// We appreciate the comment and have made the following changes:

 . we have added Figure 1 to the text.

 . we have improved the description of the related work.

 . we have reorganized the works presented in the discussion section and we have moved them into the introduction section.

• The lack of reference to important studies in this context shows a lack of a comprehensive review, which leads to some bold claims. For instance, how does this research build upon: 

Awad, Susanne F., et al. "Forecasting the burden of type 2 diabetes mellitus in Qatar to 2050: a novel modeling approach." Diabetes research and clinical practice 137 (2018): 100-108. 

or Khanal, Saval, et al. "Forecasting the amount and cost of medicine to treat type 2 diabetes mellitus in Nepal using knowledge on medicine usage from a developed country." Journal of Pharmaceutical Health Services Research 10.1 (2019): 91-99. 

Please also review the English and eventual translations like "Data transformación" and "Data Pre-Datacasting". Check also the comas and points issue: "US$ 1,125 and US$ 1.234"

R:// We appreciate the comment and have updated the papers described and checked the typos .

• Also, format the tables in the same fashion throughout the text for uniformity.

R:// We have used the same format for tables in the whole document. 

• There must be at least a reference to support the choice of performance evaluation. A few graphs on the Descriptive Analysis are a critical addition to better categorize de distributions.

R:// We have added a reference of the metrics used. 

• For the above-mentioned reasons, I believe that this manuscript has some potential but needs a comprehensive amount of work before publication.

R:// We appreciate this comment for letting us improve our paper.

Reviewer #2: Summary

The goal of this work is to assess the effectiveness of different machine learning models in estimating pharmaceutical and non-pharmaceutical expenditures associated with Diabetes, considering the comorbidities of the patient. The models considered were Linear Regression, Lasso Regression, and Neural Networks. The results showed greater accuracy in predicting pharmaceutical and non-pharmaceutical expenditures through Neural Networks.

R:// We appreciate this comment for letting us improve our paper.

Minor comments

• The submitted document had two different formatting types. In future submissions, try to uniformize the font and size used.

R:// We appreciate this suggestion, we have checked and changed the entire document using the same type of font and size.

• In table 3, the results of the linear regression display the estimation of the intercept (constant), the dummy variable for Man (M), and the dummy variable for Woman (W). This table should be reviewed, since in the linear model, including both categories and the intercept would mean we are violating the absence of perfect collinearity assumption, which means we could not estimate the model through OLS.

R:// We appreciate this comment, we have updated the tables clarifying the changes made to guarantee the assumption of the OLS model.

• More details on the methods used should be added in the Methodology and Results section. The results could also be complemented with some plots.

Major comments

Overall, this work is well written. However, it falls short concerning the implemented methodology. The result that Neural Networks are more efficient is expected. These algorithms have been used before to predict not only the risk/probability of a specific disease but also medical costs. The major contribution is the inclusion of Charle's Index as an attribute in the models. Thus, this work should be enriched with other methods, besides the ones implemented. For example, cluster the individuals based on the Charle's Index and then apply the models to each cluster or consider the panel data structure (since in this work each time that a patient is admitted to the hospital is treated as a single source of data, therefore patients with multiple registrations to the hospital in different periods are analyzed as different record).

R:// We agree with reviewers, nevertheless, we have a hard constraint given the database that doesn´t allow us to split data into clusters because we don´t have enough records and we’re not able to get more of them.

Reviewer #3: 

• The authors need to explain in depth all the used variables in the model. Note that using the same variable as independent and dependent invalidates any accuracy of the model, hence the authors must strongly justify why they used hospitalization costs as independent variable (which include pharmaceutical and non-pharmaceutical expenditures), and as dependent variable also expenditure (composed of pharmaceutical and non-pharmaceutical expenditures). This intorduces a high risk of biased resuts.

R:// We agree with the reviewer, however, we have clarified in the text that the independent variables calculated are related to the number of medicine consumption (pharmaceutical and non-pharmaceuticals) but not to the expenditures.

• The authors must also rewrite the introduction and include the study contributions, as well in the methodology, it is fundamental to support theoretically each used item/variable included in the model. It is of most importance the conclusions contain implications and limitations of the study.

R:// We agree with the reviewer, therefore we have improved the introduction including related works and we have included the implications and limitations of the study.

Kind Regards,

Javier González Rodriguez

Correspondence Author

---

## [Decision Letter · Decision Letter 1]

6 Jul 2023

PONE-D-22-29157R1Prediction of pharmaceutical and non-pharmaceutical expenditures associated with Diabetes Mellitus type II based on clinical riskPLOS ONE

Dear Dr. González R,

Thank you for submitting your manuscript to PLOS ONE. After careful consideration, we feel that it has merit but does not fully meet PLOS ONE’s publication criteria as it currently stands. Therefore, we invite you to submit a revised version of the manuscript that addresses the points raised during the review process.

We look forward to receiving your revised manuscript.

Kind regards,

Bruno Miguel Pinto Damásio

Academic Editor

PLOS ONE

Journal Requirements:

Additional Editor Comments:

Dear Dr. Gonzalez

I have received the referee reports and can now make a decision.

One of the referees is happy with the review and recommends acceptance. Unfortunately another referee recommends a major revision.

From my side, I think your paper has clear potential to be published in PLOS ONE but not in its current state, I am asking you a minor revision.

I invite you to respond point by point to the referee's report and resubmit a new version.

Kind regards,

Bruno Damásio

Reviewers' comments:

Reviewer's Responses to Questions

**Comments to the Author**

1. If the authors have adequately addressed your comments raised in a previous round of review and you feel that this manuscript is now acceptable for publication, you may indicate that here to bypass the “Comments to the Author” section, enter your conflict of interest statement in the “Confidential to Editor” section, and submit your "Accept" recommendation.

Reviewer #2: All comments have been addressed

Reviewer #4: All comments have been addressed

2. Is the manuscript technically sound, and do the data support the conclusions?

Reviewer #2: Yes

Reviewer #4: Yes

3. Has the statistical analysis been performed appropriately and rigorously? 

Reviewer #2: Yes

Reviewer #4: I Don't Know

4. Have the authors made all data underlying the findings in their manuscript fully available?

Reviewer #2: Yes

Reviewer #4: Yes

5. Is the manuscript presented in an intelligible fashion and written in standard English?

Reviewer #2: Yes

Reviewer #4: Yes

6. Review Comments to the Author

Reviewer #2: (No Response)

Reviewer #4: The authors have successfully defended all comments of previous reviewers. They have actually improved the manuscript according to the suggestions. Hence, the current form of the paper may be accepted.

They may cite some related and recent papers in the final manuscript:

https://doi.org/10.1007/s11042-020-10242-8;
https://doi.org/10.1007/978-981-13-1280-9

7. PLOS authors have the option to publish the peer review history of their article (what does this mean?). If published, this will include your full peer review and any attached files.

Reviewer #2: No

Reviewer #4: No

---

## [Author Response · Author response to Decision Letter 1]

17 Jan 2024

We are currently sending a revised manuscript, labeled 'Revised Manuscript with Track Changes-14-08-23', an unchanged manuscript labeled "Manuscript" and finally a rebuttal letter that responds.

---

## [Decision Letter · Decision Letter 2]

24 Mar 2024

Prediction of pharmaceutical and non-pharmaceutical expenditures associated with Diabetes Mellitus type II based on clinical risk

PONE-D-22-29157R2

Dear Dr. González R,

We’re pleased to inform you that your manuscript has been judged scientifically suitable for publication and will be formally accepted for publication once it meets all outstanding technical requirements.

Kind regards,

Giovanni Dolci

Academic Editor

PLOS ONE

Additional Editor Comments (optional):

Reviewers' comments:

Reviewer's Responses to Questions

**Comments to the Author**

1. If the authors have adequately addressed your comments raised in a previous round of review and you feel that this manuscript is now acceptable for publication, you may indicate that here to bypass the “Comments to the Author” section, enter your conflict of interest statement in the “Confidential to Editor” section, and submit your "Accept" recommendation.

Reviewer #2: All comments have been addressed

Reviewer #4: All comments have been addressed

2. Is the manuscript technically sound, and do the data support the conclusions?

Reviewer #2: Yes

Reviewer #4: Yes

3. Has the statistical analysis been performed appropriately and rigorously? 

Reviewer #2: Yes

Reviewer #4: Yes

4. Have the authors made all data underlying the findings in their manuscript fully available?

Reviewer #2: Yes

Reviewer #4: Yes

5. Is the manuscript presented in an intelligible fashion and written in standard English?

Reviewer #2: Yes

Reviewer #4: Yes

6. Review Comments to the Author

Reviewer #2: Just a small detail: on line 366 it should be "absence of perfect collinearity ...", instead of "perfect collinearity".

The rest of the comments were addressed.

Reviewer #4: I think that the authors have adequately addressed the comments made by the reviewers in the revised version of the manuscript. Therefore, I have no further comments.

7. PLOS authors have the option to publish the peer review history of their article (what does this mean?). If published, this will include your full peer review and any attached files.

Reviewer #2: No

Reviewer #4: No
